# Altered mRNA and Protein Expression of Monocarboxylate Transporter MCT1 in the Cerebral Cortex and Cerebellum of Prion Protein Knockout Mice

**DOI:** 10.3390/ijms22041566

**Published:** 2021-02-04

**Authors:** Sanja Ramljak, Matthias Schmitz, Cendrine Repond, Inga Zerr, Luc Pellerin

**Affiliations:** 1Digital Diagnostics AG, 55129 Mainz, Germany; 2Department of Neurology, University Medicine Goettingen and The German Center for Neurodegenerative Diseases (DZNE), 37075 Goettingen, Germany; ingazerr@med.uni-goettingen.de; 3Département de Physiologie, Université de Lausanne, 1005 Lausanne, Switzerland; Cendrine.Repond@unil.ch (C.R.); luc.pellerin@univ-poitiers.fr (L.P.); 4Centre de Résonance Magnétique des Systèmes Biologiques, UMR5536 CNRS, LabEx TRAIL-IBIO, Université de Bordeaux, 33760 Bordeaux CEDEX, France

**Keywords:** cellular prion protein, prionprotein knockout, MCT1, MCT4, Na^+^/K^+^ ATPase, cortex, qRT-PCR, Western blot

## Abstract

The effect of a cellular prion protein (PrP^c^) deficiency on neuroenergetics was primarily analyzed via surveying the expression of genes specifically involved in lactate/pyruvate metabolism, such as monocarboxylate transporters (*MCT1*, *MCT2*, *MCT4*). The aim of the present study was to elucidate a potential involvement of PrP^c^ in the regulation of energy metabolism in different brain regions. By using quantitative real-time polymerase chain reaction (qRT-PCR), we observed a marked reduction in MCT1 mRNA expression in the cortex of symptomatic Zürich I *Prnp^−/−^* mice, as compared to their wild-type (WT) counterparts. MCT1 downregulation in the cortex was accompanied with significantly decreased expression of the MCT1 functional interplayer, the Na^+^/K^+^ ATPase α2 subunit. Conversely, the MCT1 mRNA level was significantly raised in the cerebellum of *Prnp^−/−^* vs. WT control group, without a substantial change in the Na^+^/K^+^ ATPase α2 subunit expression. To validate the observed mRNA findings, we confirmed the observed change in MCT1 mRNA expression level in the cortex at the protein level. MCT4, highly expressed in tissues that rely on glycolysis as an energy source, exhibited a significant reduction in the hippocampus of *Prnp^−/−^* vs. WT mice. The present study demonstrates that a lack of PrP^c^ leads to altered MCT1 and MCT4 mRNA/protein expression in different brain regions of *Prnp^−/−^* vs. WT mice. Our findings provide evidence that PrP^c^ might affect the monocarboxylate intercellular transport, which needs to be confirmed in further studies.

## 1. Introduction

PrP^c^is associated with heterologous biological processes. In the last few years, various reports have indicated the involvement of PrP^c^ in the maintenance of glucose homeostasis [1,2,3,4,5]. Besides this, an effect of PrP^c^ on glycolysis was suggested in several earlier studies [6,7,8]. A recent metabolomic analysis demonstrated reduced glycolysis and glucose utilization in the hippocampus and the cortex of prion-diseased mice [2]. Moreover, major alternations in mitochondrial metabolism were identified in the brain of sporadic Creutzfeldt–Jakob disease (sCJD) patients [9], which may be a consequence of PrP^c^ loss of function.

Deficits in glucose availability and disruption of mitochondrial functions were also seen in Alzheimer’s disease (AD) patients [10]. Furthermore, an impairment of glycolysis was demonstrated to sensitize human astrocytes to amyloid beta toxicity and to induce amyloid aggregation [11]. Another recent study reported a decreased expression of monocarboxylate transporters (MCT1, MCT2, MCT4) and lactate dehydrogenase A and B (LDH-A and LDH-B) in the brains of double transgenic amyloid precursor protein/presenilin 1 (APP/PS1) mouse model of AD [12]. MCTs are membrane-bound proteins with body-wide distribution, which enable the intercellular shuttling of energy metabolites such as lactate and pyruvate, whereas LDH isoenzymes enable the reversible conversion of pyruvate to lactate and thus play a critical role in glycolytic metabolism. Earlier, we showed that PrP^c^ overexpression enhances MCT1 protein level in HEK 293 cells [13]. Interestingly, we also found that LDH-A and LDH-B protein expression levels were markedly increased in the brain of wild-type (WT) vs. Prnp knockout mice when exposed to ischemic (stress) conditions, but exhibited invariable expression under non-ischemic conditions [13]. The latter finding is significant in the light of a well-known fact that PrP^c^ mice exhibit a higher tolerance toward different stress conditions as compared with *Prnp^−/−^* mice [14,15].

Lately, MCT1 mRNA levels have been shown to be markedly reduced in the frontal cortex of CJD MM1 and VV2 patients [16]. The cerebral cortex together with the cerebellum is one of the most frequently affected brain regions in CJD patients [17,18,19].

Therefore, we aimed to investigate if the presence/absence of PrP^c^ in the cortex and the cerebellum of WT and *Prnp^−/−^* mice may directly influence MCT mRNA/protein expression levels along with additional genes involved in lactate/pyruvate energy metabolism, such as *basigin*, *LDH-A* and *LDH-B*. In particular, the former protein is of interest because it is recognized as necessary for correct MCT1 localization and functioning [20]. Moreover, we investigated the hippocampus, in which important glucose metabolism changes were reported in prion-diseased mice [2].

## 2. Results

As to 3-month-old mice, 9-monthold *Prnp^−/−^* (*Zurich I*) mice showed various impairments during aging (appearing after 9 months) in comparison to WT mice (Appendix A) [21]. Young animals (e.g., after 3 months) exhibited no significant behavioral differences. Therefore, we expected the potential changes between both groups at a later time point. Animals older than 12–14 months generally showed age-related deficits (also in WT mice). That is why we considered 9 months of age as an optimal time point for our study.

In order to investigate if *MCT1*, *MCT2* and *MCT4* expression levels are modified in *Prnp^−/−^* vs. WT mice, we employed qRT-PCR. The qRT-PCR results showed a marked four-fold downregulation of *MCT1* mRNA expression in the cerebral cortex of *Prnp^−/−^* as compared to WT mice (*p* = 0.0043) (Figure 1A). This finding was reproduced at the protein level with an approximately two- to three-fold decrease in MCT1 expression in the cerebral cortex of *Prnp*^−/−^ mice vs. WT mice after 9 months (Figure 2, *p* < 0.001), but not after 3 months of age (Appendix A). Likewise, a small but significant decline in *Na^+^/K^+^ ATP-ase α2* subunit mRNA expression was observed in the same brain region of the *Prnp^−/−^* mice (*p* = 0.0444) (Figure 1B). No changes were observed in the mRNA expressions of *MCT2* and *MCT4* (Figure 1C,D).

On the contrary, *MCT1* was upregulated in the cerebellum of *Prnp^−/−^* vs. WT group (*p* = 0.0068) (Figure 3A) without a significant change in *Na^+/^K^+^ ATP-ase α2* subunit expression (Figure 3B). No significant differences in *MCT2* expression were detected between both experimental groups. However, a recognizable trend in *MCT2* expression is apparent in the cerebellum, i.e., an increase in mRNA expression in the *Prnp^−/−^* vs. WT group, which parallels the expressional increase in MCT1 in the cerebellum (Figure 3C). No change in *MCT4* expression could be detected in the cerebellum (Figure 3D).

In the hippocampus, no significant changes in expression could be found for *MCT1* and *MCT2* (Figure 4A,B). However, *MCT4* exhibited an altered mRNA expression between the two experimental groups, in which there was a slight, but significant, *MCT4* decrease found in *Prnp^−/−^* mice, as compared to their counterparts (*p* = 0.0366) (Figure 4C). No changes were noticed in the mRNA expression of *basigin*, *LDH-A* or *LDH-B* in any of the three brain regions tested (data not shown).

## 3. Discussion

Up to now, the understanding of PrP^c^’s role in neuroenergetic processes has been limited. Therefore, we examined if a lack of PrP^c^ may modulate the expression of genes known to be critically involved in supporting unobstructed glycolytic processes that are of utmost importance, especially under stress conditions when energy demand is high.

As a model we chose 9-month-old WT and *Prnp^−/−^* mice because of aging-related behavioral changes in *Prnp^−/−^* mice starting at this age [21], and we propose an association between behavioral deficits and the energy metabolism.

In the present study, we found that the mRNA expression levels of MCT1 and MCT4 as well as Na^+^/K^+^-ATPase α2 subunit are considerably changed in either the cortex, the cerebellum or the hippocampus of *Prnp^−/−^* vs. WT mice. Additionally, we showed that MCT1 protein levels in the cortex are upregulated in 9-month-old WT mice as compared to *Prnp^−/−^* mice (behavioral changes between both groups obvious in respect to learning, anxiety, curiosity), but not in 3-month-old mice (no behavioral changes between both groups obvious). This finding hints at the association of the presence/absence of PrP^c^, changes in MCTs mRNA/protein expression and potential impact on neuroenergetics.

Reduced *PRNP* mRNA expression levels were previously observed in the frontal cortex of sCJD patients [22], where the cellular function of PrP was lost due its conversion to PrP scrapie, and the abnormal imaging patterns detectable in the cerebral cortex of CJD-affected patients are well characterized [23]. MCT1 mRNA expression and immunoreactivity were identified as markedly diminished in the same brain area of sCJD as compared to control patients [16]. Therefore, investigating the differential cortical mRNA expression of genes engaged in neuroenergetics seemed to be a relevant issue.

A four-fold downregulation of MCT1 mRNA and protein expression, observed in the present study in the cortex of *Prnp^−/−^* mice vs. WT mice, suggests the inadequate intercellular transport of lactate and/or pyruvate, which may be a consequence of a lack of PrP^c^. This finding is in concordance with the above-mentioned reduced MCT1 cortical mRNA and protein expression shown in sCJD patients [16]. Another report demonstrated that the downregulation of MCT1 in glioma cells results in decreased lactate production [24]. The knockdown of MCT1 results in aberrant axon morphology and neuronal death, a phenotype that could be rescued by supplying lactate [25]. Important disturbances in lactate metabolism were previously detected in the brain of *Prnp* knockout mice [26].

Kleene at al. 2007 [27] stated that PrP^c^ regulates lactate transport via MCT1 in cultured astrocytes conjointly with Na^+^/K^+^ α2/β2 ATPase and basigin, and that the activity of the former is reduced in PrP-deficient astrocytes. We observed significantly lowered Na^+^/K^+^ ATPase α2 subunit mRNA expression levels in *Prnp^−/−^* vs. WT mice, and no difference in the expression of basigin that directly interacts with the Na^+^/K^+^ ATPase α2 subunit. Glutamate uptake is known to stimulate aerobic glycolysis in astrocytes by activating the α2 subunit of the Na^+^/K^+^ ATPase [28]. A two-fold reduction in neuronal and astroglial Na^+^-dependent glutamate uptake was detected in the absence of PrP^c^ [29].

Interestingly, MCT1 was slightly but significantly elevated in the cerebellum of Prnp^−/−^ mice and a marked upward trend was seen in the MCT2 expression.

MCT4, another monocarboxylate proton-coupled transporter, showed slight but significant downregulation in the hippocampus of *Prnp^−/−^* vs. WT mice, suggesting an impaired lactate export from astrocytes to neurons. Disrupting the expression of MCT1 or MCT4 (glial transporters), and/or MCT2 (neuronal transporter), in the rat hippocampus prevented learning and long-term memory formation. The latter deficiency could be rescued by exogenously provided lactate [30], underlining the importance of lactate shuttling from astrocytes to neurons. Amongst the most prominent phenotypes of *Prnp^−/−^* mice are cognitive deficits and memory impairment [31]. Astrocytes are known to export lactate either through MCT1 or MCT4 to power oxidative neurons expressing MCT2 [32]. A bidirectional impact of the activity between cerebellum and hippocampus has been demonstrated [33]. Although the role of the cerebellum in cognitive functions is still controversial, there is growing evidence of its contributions to cognitive functions, such as attention, language, working memory, and visuospatial navigation [34,35]. Therefore, it is conceivable that all three brain structures cooperate in a compensatory way at the energetic level, and therefore show differential up-/downregulation of MCTs as compared to each other.

No differential regulation of LDH-A or LDH-B could be evidenced between the two experimental groups in the present study, which is in concordance with earlier results [13] that showed that the expression levels of both LDH isoenzymes increase in WT vs. *Prnp^−/−^* mice solely following hypoxic injury.

Investigating the expression levels of MCTs and other genes important for the optimal execution of neuroenergetic processes in WT vs. *Prnp^−/−^* mice after exposure to hypoxia (stress condition) appears to be a proper approach to verifying if the increased infarction volumes observed in *Prnp^−/−^* mice may be a consequence of the disturbed expression of MCTs and glycolysis. As such, PrP^c^ emerges as a novel protein that may prove importantance for the regulation of MCT1 and MCT4 expression, and its lack or loss of function may thereby accelerate the process of neurodegeneration.

## 4. Material and Methods

### 4.1. Animals

Zürich I *Prnp^−/−^* mice of the C57BL/6J genetic background were generated as previously described by *Bueler* et al. [36]. As controls, we used WT mice from the same genetic background. The body weight of all animals varied between 25 and 35 g and they were either 3 or 9 months-old. *Prnp^−/−^* mice showed age-dependent behavioral abnormalities as given in the Appendix A and and earlier reports [21,37]. Five (WT group) and six (*Prnp^−/−^* group) animals per experimental group were tested for differential mRNA expression levels.

### 4.2. Brain Homogenates for RNA Preparation

After euthanization by cervical dislocation, mouse brains were dissected on ice and collected cortices were lysed and homogenized in 350 μL lysis buffer (RLT Buffer, Qiagen) using the VWR Pellet mixer (VWR, Dietikon, Switzerland) according to the manufacturer’s instructions. Total RNA was isolated on spin columns with silica-based membranes (RNeasy Mini Kit, Qiagen, Basel, Switzerland), following the manufacturer’s instructions. RNA was eluted with 30 μL of H_2_O. Two hundred ng of purified RNA was reverse transcribed in a volume of 50 μL using the RT High Capacity RNA-to-cDNA Kit (Applied Biosystems, Rotkreuz, Switzerland). Quantitative real-time PCR analysis was performed on cDNA obtained with the Applied Biosystems ViiA^TM^7 (Applied Biosystems, Rotkreuz, Switzerland) Real-Time PCR System using Power SYBR Green Taq polymerase master mix (Applied Biosystems, Rotkreuz, Switzerland). Primer sequences used for mRNA quantification were directed against *LDH-A*, *LDH-B*, *basigin*, *Na^+^/K^+^ATPase α2* subunit, *MCT1*, *MCT2* and *MCT4* mRNAs, as well as *β-Actin* mRNA used as an endogenous control (See Table 1 for sequences). For data analysis, the raw threshold cycle (CT) value was first normalized to the endogenous control for each sample to obtain the ^Δ^CT value. The normalized ^Δ^CT value was then calibrated to the control cell samples to obtain the 2^−^^ΔΔCT^.

### 4.3. Primer Sequence

The list of all forward and reverse sequences used for qRT-PCR experiments is provided in Table 1.

### 4.4. Protein Analysis

#### Brain Homogenates

After euthanization by cervical dislocation, mouse brains were dissected on ice into three different brain regions: hippocampus (H), cortex (C), cerebellum (CB). Samples were homogenized in homogenization buffer (50 mM Tris HCl (pH 7.5), 150 mM NaCl, 2 mM EDTA, 1% Triton X-100 and protease inhibitors), sonicated for 5 min in a water bath and incubated with rotation for 15 min. Insoluble debris was removed by a centrifugation step for 20 min at 13,000× *g* at 4 °C. The supernatant was transferred to a separate tube and stored at −80 °C.

### 4.5. Western Blotting

For Western blot analysis we used the monoclonal anti-MCT1 (Abcam, Cambridge, UK) diluted 1:500, anti-PrP antibody SAF70 (SPI-Bio, Montigny Le Bretonneux, Paris, France) diluted 1:500, and monoclonal GAPDH antibody diluted 1:5000 (Abcam, Cambridge, UK). We followed a standard protocol published before [21]. After Western blotting detection, bands were analyzed via Chemi-Doc (Bio-Rad, Munich, Germany). A densitometric analysis of band intensities was performed with *Image Lab* software version 6.0.1.

### 4.6. Ethics Approval

The study was approved by the Lower Saxony State Office for Consumer Protection and Food Safety (No. 16/2073).

### 4.7. Statistical Analysis

Results are presented as means ± SEM (standard error of the mean). Statistical analysis was performed using *GraphPad Prism* 8 software. Normality was tested with the *Kolmogorov–Smirnov* test. Depending on the result of the normality test, an unpaired Student’s *t*-test or an unpaired *t*-test with *Welch’s* correction (when equal variance was not assumed) was used. Moreover, we performed an animal experimentation sample size calculation to estimate the number of animals required to attain statistical significance. Significant differences are indicated for *p*-values of <0.05, <0.01, or <0.001. All qRT-PCR and western blotting experiments were performed in triplicates.

## Figures and Tables

**Figure 1 ijms-22-01566-f001:**
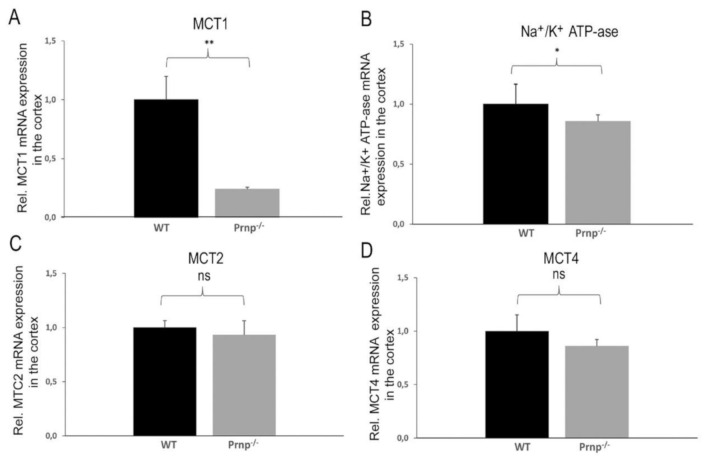
(**A**–**D**). Differential mRNA expression levels of monocarboxylate transporters *MCT1*, *MCT2*, *MCT4* and *Na^+^/K^+^ ATP-ase α2* subunit in the cortex of WT and Prnp^−/−^ mice. qRT-PCR analysis revealed a significant decrease in *MCT1* and *Na^+^/K^+^ ATP-ase α2* subunit expression in the cortex tissue of *Prnp^−/−^* as compared to a control group. Sample size of WT group: *n* = 5; *Prnp^−/−^* group: *n* = 6. Displayed are means ± SEM. A *p*-value < 0.01 is considered as very significant (**), <0.05 as significant (*) and ≥0.05 as not significant (ns).

**Figure 2 ijms-22-01566-f002:**
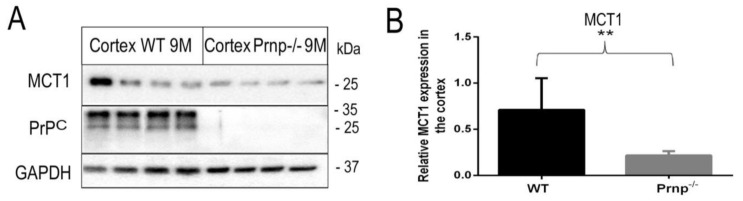
Western blot and densitometric analysis of MCT1 and PrP^c^ expression in the cortex of 9-month-old WT and *Prnp^−/−^* mice. (**A**) Homogenates prepared from cortex were examined for MCT1 and PrP^c^ expression by Western blotting. All brain homogenates of WT mice exhibited PrP^c^ expression (diglycosylated at 35 kDa, monoglycosylated at 33 kDa and unglycosylated at 26 kDa), whereas *Prnp^−/−^* mice blots confirmed the absence of PrP^c^. An equal protein load (20 µg per lane) is shown by GAPDH expression. (**B**) Densitometric quantification of band intensities via *Image Lab 6.0.1* revealed a higher protein expression of MCT1 in WT as compared to *Prnp^−/−^* mice. The analyses were performed on *n* = 4 animals per group in three different Western blots. Displayed are means ± SEM. A *p*-value < 0.01 is considered as very significant (**).

**Figure 3 ijms-22-01566-f003:**
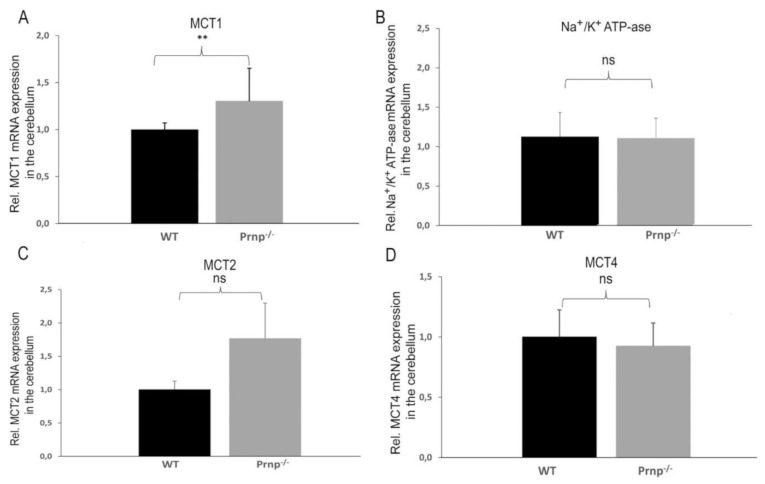
(**A**–**D**). Differential mRNA expression of *MCT1*, *MCT2*, *MCT4* and *Na^+^/K^+^ ATP-ase α2* subunit in the cerebellum of WT and *Prnp^−/−^* mice. The mRNA expression level of MCT1 was significantly increased in the cerebellum of *Prnp^−/−^* vs. a control group, while *MCT2*, *MCT4* and *Na^+^/K^+^ ATP-ase α2* subunit expression remained unchanged. Sample size of WT group: *n* = 5; *Prnp^−/−^* group: *n* = 6. Displayed are means ± SEM. A *p*-value < 0.01 is considered as very significant (**) and ≥0.05 as not significant (ns).

**Figure 4 ijms-22-01566-f004:**
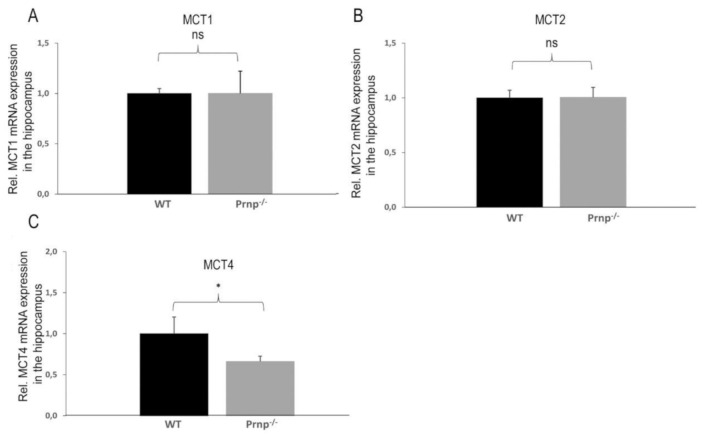
(**A**–**C**). Differential mRNA expression of *MCT1*, *MCT2* and *MCT4* in the hippocampus of WT and *Prnp^−/−^* mice. Quantification of the mRNA expression revealed a lower *MCT4* expression in the hippocampus of the *Prnp^−/−^* experimental group than in control group, while at the same time levels of *MCT1* and *MCT2* were unaltered. Sample size of WT group: *n* = 5; *Prnp^−/−^* group: *n* = 6. Displayed are means ± SEM. A *p*-value < 0.05 is considered as significant (*) and ≥0.05 as not significant (ns).

**Table 1 ijms-22-01566-t001:** List of primer sequences used in qRT-PCR experiments.

Name	Forward Sequence	Reverse Sequence
*LDH-A*	CAGTGGCTTTGCCAAAAACCGAGT	CCATCAGGTAACGGAACCGCG
*LDH-B*	CCTGCTGACTTTGCAGTGGCTCC	TCGCCGCGGCAGCCTCATCAT
*Basigin*	CAAGGTACTGCAGGAGGACACTCT	TCAGGAAGGAAGATGCAGGAATATT
*Na^+^/K^+^-ATPase α2*	GAGACGCGCAATATCTGTTTCTT	ACCTGTGGCAATCACAATGC
*MCT1*	TTGGACCCCAGAGGTTCTCC	AGGCGGCCTAAAAGTGGTG
*MCT2*	CAGCAACAGCGTGATAGAGCTT	TGGTTGCAGGTTGAATGCTAAT
*MCT4*	CGGCTGGCGGTAACAGAGTA	CGGCCTCGGACCTGAGTATT
*β-Actin*	GCTTCTTTGCAGCTCCTTCGT	ATATCGTCATCCATGGCGAAC

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
