# Peer review of "Altered mRNA and Protein Expression of Monocarboxylate Transporter MCT1 in the Cerebral Cortex and Cerebellum of Prion Protein Knockout Mice"

_ijms, 2021, doi:10.3390/ijms22041566_

Round 1

Reviewer 1 Report

In this revised version of a brief report, Ramljak and colleagues still aim to assess the involvement of PrPc in the regulation of energy metabolism in different brain regions using PrP-/- and WT mice. They planned to investigate if the presence or absence of PrPc in the cortex and cerebellum 
of WT and PrP-/- mice may directly influence MCT mRNA expression levels along with additional genes involved in lactate / pyruvate energy metabolism. 

Under the experimental conditions used in this study, the authors observed a marked reduction of MCT1 mRNA expression in the cortex of symptomatic Zurich 1 Prnp-/- mice in comparison with the WT mice. Conversely, the authors state that MCT1 mRNA level was significantly raised in the cerebellum of 
Prnp-/- vs. WT control mice. MCT4 highly expressed in tissues that rely on glycolysis as an energy 
source exhibited a significant decline in the hippocampus of Prnp-/- vs. WT mice. 

Based on the data obtained by the authors, they suggest that lack of PrPc leads to altered MCT1 and MCT4 mRNA expression in different brain regions of Prnp-/- versus WT mice. The authors go on to conclude that their data suggest that PrPc might affect the monocarboxylate intercellular transport. However, this finding will need to be confirmed in further studies.

Main Points and Comments:

  1. The report is much better written, reasonably easy to read and presents some nice slightly more comprehensive data than in the original version of the paper.
  2. The abstract has had 3 new sentences added and the title has changed although most of the paper is the same as it was in the original version.
  3. I have worked through all the author’s comments and replies to both the Academic Editor and my previous review and I am happy that the authors have attempted to give reasonably robust answers or suggestions for most of the points that were raised initially. The paper has had a new Figure 4 added and also Supplementary Table 1 and Supplementary Figure 1 as well as an extra Figure 2B panel.
  4. I agree that this is a “novel pilot study” and as such it is now in a better format.
  5. Can the authors explain why they have changed the keywords with the addition of MCT4 but this is no longer in the title?
  6. The addition of the western blots to demonstrate protein expression levels is useful. Please can the molecular weight markers be added to Figure 4A and also to Supplementary Figure 1A?
  7. Can the authors show slightly more of the tracks in the PrPc western blots? WT mice should demonstrate 3 distinct bands of PrP and from the images shown I cannot tell whether the un-glycosylated PrPc has been cut off the bottom or whether the di-glycosylated and mono-glycosylated PrP bands are just squashed together.
  8. I am concerned about the lack of ethical approval stated at the end of the paper. Please can this be included?

This revised version is an improvement on the original although not all the points have been completely fully addressed. I agree that the author’s have made an attempt to clarify or explain the issues and so I am happy that this version is much better presented than the original.

There are a few sections in the new paper that have been changed that were not highlighted by the authors (such as lines 17-18, 126-127, 155-156), but this resubmitted version with the additions and modifications have made a huge improvement to the report.

Author Response

Response

  1. The report is much better written, reasonably easy to read and presents some nice slightly more comprehensive data than in the original version of the paper.
  2. The abstract has had 3 new sentences added and the title has changed although most of the paper is the same as it was in the original version.
  3. I have worked through all the author’s comments and replies to both the Academic Editor and my previous review and I am happy that the authors have attempted to give reasonably robust answers or suggestions for most of the points that were raised initially. The paper has had a new Figure 4 added and also Supplementary Table 1 and Supplementary Figure 1 as well as an extra Figure 2B panel.
  4. I agree that this is a “novel pilot study” and as such it is now in a better format.
  5. Can the authors explain why they have changed the keywords with the addition of MCT4 but this is no longer in the title?

Thank you very much for your helpful comments.

 Point 5.) The authors introduced two new keywords i.e. „Western blot“ and „MCT4“. The introduction of the word „Western blot“ refers to the Western blot analysis in the new version of the manuscript. The authors also included the keyword „MCT4“ in addition to MCT1 and Na+/K+ ATP-ase because MCT4 also showed mRNA expressional changes (in the hippocampus) as MCT1 and Na+/K+ ATP-ase did in other brain regions. There is no reason to exclude this one. However, to include Na+/K+ ATP-ase and MCT4 in the title would make the title of the manuscript too long and the findings in respect to MCT1 mRNA/protein expressional changes are the most important ones.

6. The addition of the western blots to demonstrate protein expression levels is useful. Please can the molecular weight markers be added to Figure 4A and also to Supplementary Figure 1A?

7. Can the authors show slightly more of the tracks in the PrPc western blots? WT mice should demonstrate 3 distinct bands of PrP and from the images shown I cannot tell whether the un-glycosylated PrPc has been cut off the bottom or whether the di-glycosylated and mono-glycosylated PrP bands are just squashed together.

Point 6 and 7) We have revised Figure 4A and the Supplementary Figure 1A according to the suggestions. We added the molecular weight, showed slightly more of the tracks in the PrPc western blots to proof that no PrP band has been missed and we stated in the legend the molecular weight of all three glycoforms of PrPC. Due to their equal molecular weights of 35 and 33 kDa the discrimination between a weak mono and a dominant diglycosylated PrP isoform is only barely recognizable due to the composition of the acrylamide gel.

I am concerned about the lack of ethical approval stated at the end of the paper. Please can this be included?

Point 8) At the end of the method part we added the ethical approval of the study.

Reviewer 2 Report

Thank you for submitting a corrected draft of the manuscript titled "Altered mRNA and protein expression of monocarboxylate transporter MCT1 in the cerebral cortex and cerebellum of prion protein knockout mice" to International Journal of Molecular Sciences. I appreciate the time and effort that authors have put into providing responses to the comments and suggestions to improve the manuscript.

In my view, the incorporated changes, edits, and responses address most of the issues and concerns that described in the first review. Therefore, I think the manuscript can be proceed for publication in the current form.

Author Response

Thank you very much for your helpful comments and the kind appreciation.